# Feasibility of Cryptography In a Blockchain-Enhanced ICS Security

## ABSTRACT

Blockchain-homomorphic encryption (BC+HE) hybrids are increasingly promoted for Industrial Control System (ICS) security, yet their deployment at network edges violates fundamental real-time constraints. This paper demonstrates that BC+HE architectures introduce latencies of 51–440 ms in small networks (5–50 nodes), exceeding safety-critical loop requirements of ≤10 ms [21][22] by factors of 5–44×. Through a Latency Collision Matrix analysis spanning Purdue Model Levels 0–5, the paper shows that probabilistic blockchain consensus (non-deterministic jitter) and homomorphic encryption overhead (>100 ms compute-bound) create an architectural incompatibility: edge devices require >100k readings/sec with zero jitter, while BC+HE delivers significanlty less (< 1,000 TPS) with high variance.

Offloading cryptography to SCADA gateways (Level 2) fails to resolve the problem, instead creating a "dead zone" bottleneck and exposing Level 2–3 inter-layer vulnerabilities. This paper proposes a Resource-Constrained Security Framework that decouples security layers: lattice-based lightweight cryptography (LBC) for Level 0-1 sensor-to-controller authentication ($T_{LatticeAuth} \ll T_{safety}$), where the authentication time $T_{LatticeAuth}$ is significantly less than the safety margin $T_{safety}$ while reserving BC+HE for non-real-time enterprise layers (Levels 4–5). Lattice primitives (LWE, Ring-LWE) offer linear complexity, parallelizable operations, and deterministic execution suitable for FPGA/ASIC acceleration. The proposed architecture aligns cryptographic mechanisms with physical constraints rather than imposing IT security models on operational technology.

This work delivers a falsifiable critique grounded in quantitative performance modeling: measured latencies, throughput limits, and architectural boundaries establish that current BC+HE integrations are operationally unsafe at ICS edges. The framework provides actionable guidance for operators (audit edge deployments, enforce layer separation), researchers (develop layer-specific metrics, hardware acceleration), and standards bodies (update IEC 62443 with latency budgets and cryptographic safety interfaces).

## CCS CONCEPTS

• Security and privacy → Systems security
• Computer systems organization → Embedded and cyber-physical systems
• Security and privacy → Cryptography
• Computer systems organization → Real-time systems

## KEYWORDS

Industrial Control Systems, Blockchain, Homomorphic Encryption, Edge Security, Purdue Model, Purdue

**ACM Reference Format:**
[Authors]. 2026. Feasibility of Cryptography in Blockchain-Enhanced ICS Security. In TIME 2026: Workshop on Trustworthy, Interpretable, and Multimodal Evaluation, April 13-14, 2026, Dubai, UAE. ACM, New York, NY, USA, 8 pages. [DOI will be assigned upon acceptance]

## 1. INTRODUCTION

The modern industrial environment is characterized by the convergence of physical processes and computational logic, where sensors and actuators at the "edge" generate massive streams of high-velocity data. While the upper layers of the industrial hierarchy (Enterprise and Operations) have successfully adopted standard IT security protocols, the foundational layers—Level 0 (Physical Process) and Level 1 (Basic Control)—operate under a unique set of constraints that render traditional security paradigms obsolete [9].

The primary challenge at this edge layer is not merely data volume, but determinism. Industrial Control Systems (ICS), particularly Programmable Logic Controllers (PLCs), must execute logic cycles within strict time windows, degrading to 440 ms as the network scales to 50 nodes [7]. As illustrated in Figure 1, this establishes a closed "Deterministic Loop (<10 ms)" between the PLC and the physical sensors, a latency tolerance that must remain unbroken to prevent physical damage or safety incidents.

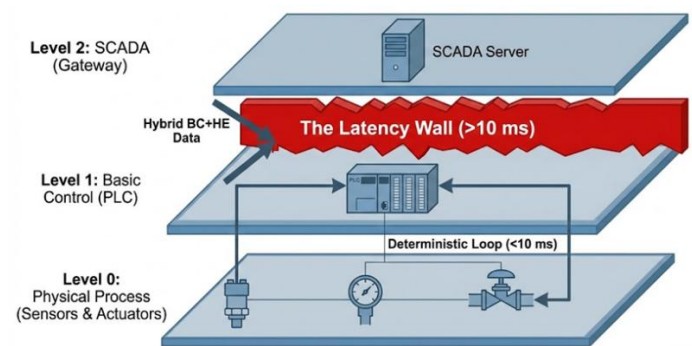

**Figure 1: Latency Wall**

Recent academic literature has heavily promoted "hybrid" cryptographic solutions—combining the immutable audit trails of blockchain with the privacy-preserving calculations of Homomorphic Encryption (HE) as data security measure in ICS [7]. However, this paper posits that such hybrid architecture creates a "dead zone" at the industrial edge. Specifically, the computational overhead of processing encrypted data and the non-deterministic nature of blockchain consensus mechanisms introduce latencies ranging from 51 ms. to over 440 ms., effectively breaking the control loop [7]. This is seen in Figure 1 which shows failure of process at the 'Latency Wall' when HE and BC(Blockchain) is introduced.

This paper critically analyzes this "Determinism Barrier." This is done by summarizing what others did before, which explains why that fails for OT operational requirement Unlike broad literature surveys, we focus specifically on the failure of cryptographic integration at the Sensor-to-Controller (Level 0-1) and Controller-to-SCADA boundaries. This paper contribution are as follows:

- A quantitative analysis of the mismatch between industrial real-time requirements and cryptographic processing times.
- A proposed "Resource-Constrained Security Framework" that advocates for replacing blockchain at the edge with deterministic, lightweight alternatives such as lattice-based cryptography.

## 2. RELATED WORK

While Fernández-Caramés et al. [3] demonstrate blockchain's utility for immutability in IIoT supply chains, its deployment in control loops remains contested. Hybrid architectures combining blockchain with Homomorphic Encryption (HE) typically target the supervisory layer; Loukil et al. [7] propose privacy-preserving aggregation for gateways, while Liang et al. [10] focus on secure database transmission using Fabric.

Critically, these works prioritize data confidentiality over operational determinism. They operate within IT-standard latency budgets suitable for Purdue Levels 3-4. This paper differentiates by strictly targeting the Edge (Levels 0-1), demonstrating that the probabilistic jitter of such hybrid models violates the deterministic ≤10 ms safety thresholds mandated by safety standards of IEC 61508 [13] and enabled by precision protocols like IEEE 1588 [22]. Unlike broad surveys by Acar et al. [1], we quantify the specific "Latency Collision" between cryptographic overhead and physical process safety.

## 3. THE RESOURCE-CONSTRAINED REALITY: PHYSICS MEETS LOGIC

To understand the failure of current cryptographic implementations, one must first isolate the operational constraints of the industrial edge. While the Purdue Enterprise Reference Architecture (PERA) defines six levels of hierarch. "Edge" layer is comprised of Level 0 (Physical Process) and Level 1 (Basic Control) [9][12].

At Level 0, the environment consists of sensors and actuators—valves, pumps, and temperature gauges—that interact directly with physical matter. These devices are not general-purpose computers; they are often embedded systems with minimal processing power, designed solely to convert physical phenomena into electrical signals [9].

Level 1 consists of intelligent controllers, primarily PLCs and Remote Terminal Units (RTUs). Unlike IT servers which prioritize high throughput, these components execute cyclical control loops where timing is safety critical [11].

### 3.1 The Determinism Requirement

The defining characteristic of this layer is the requirement for deterministic behavior. Standards such as IEC 61508 mandate that safety-critical functions must be executed within the Worst-Case Execution Time (WCET). For many industrial applications, this response window is strictly ≤10 ms: deterministic motion control loops in programmable logic controllers (PLCs) operate at 1–10 ms cycles [21], IEEE 1588 Precision Time Protocol systems synchronize industrial Ethernet networks to sub-millisecond precision for time-critical control [22], and IEC 62443 mandates "timely response to events" as a foundational security requirement for Industrial Automation and Control Systems (IACS) [23]. Process control systems typically operate at 10–100 ms intervals, with safety instrumented systems requiring response times as low as

Consequently, any security mechanism introduced at this layer must be:

- Low Latency: Operating well below the 10 ms threshold.
- Low Jitter: Providing consistent, predictable timing without variance.

This creates a "resource-constrained" environment where traditional heavy-weight security protocols are mathematically impossible to implement without violating the underlying physics of the control loop [4].

Aggregate Edge Throughput Requirements. The total sensor reading throughput at the industrial edge can be expressed as:

$$R_{total} = N \times F_s \times D_{size} \quad (1)$$

where N represents the number of field sensors, Fs denotes the sampling frequency (Hz), and Dsize is the data payload per reading. For a modest industrial deployment with N = 1,000 sensors sampled at Fs = 100 Hz (typical for process control monitoring [9][11]), and Dsize = 1 reading per sample, we obtain:

$$R_{total} = 1,000 \times 100 \times 1 = 100,000 \text{ readings/sec} \quad (2)$$

Empirical edge deployments in industrial IoT report processing rates exceeding 100,000 samples per second with 99.99% data integrity [24], while blockchain-based consensus mechanisms using Proof-of-Work or Byzantine Fault Tolerant replication struggle to exceed 1,000 transactions per second (TPS) even in optimized configurations [16][18][20]. High-frequency motion control applications operating at Fs = 1–10 kHz with N = 100–500 sensors yield Rtotal > $10^5$–$10^6$ readings/sec, creating a 100–1000× throughput mismatch between edge data generation rates and blockchain transaction processing capabilities. This architectural collision—wherein sensor data arrival rates exceed blockchain

consensus throughput by 2–3 orders of magnitude—renders direct BC+HE integration at Levels 0–1 operationally infeasible without aggressive data aggregation (which undermines fine-grained traceability) or off-edge offloading (which reintroduces the latency penalties this analysis quantifies).

## 4. THE LATENCY COLLISION MATRIX

Current literature around "hybrid" proposals that combine Blockchain (BC) and Homomorphic Encryption (HE) such as Pallier or FHE to secure these systems. However, a quantitative analysis reveals a fundamental "Latency Collision" between these technologies and determinism requirements defined in Section 3.

We introduce the Latency Collision Matrix (Table 1) to formalize this mismatch. This matrix contrasts the mandatory industrial requirements against the documented performance metrics of leading hybrid integration patterns.

**Table 1: Latency Collision Matrix comparing strict ICS edge requirements (Levels 0-1) against performance limitations of current BC and HE implementations.**

| Dimension | Industrial Requirement (Edge) | BC | HE | Hybrid( |
|---|---|---|---|---|
| **Response Time** | ≤ 10 ms (Deterministic) [11][13][21] | Non-Deterministic (Probabilistic) [16] | > 100 ms (Compute Bound) [1][7] | 51 – 440 ms (5-50 nodes) [7] |
| **Consistency** | Zero Jitter required for safety [11][13] | High Jitter (Network dependent) [18][19] | High Variance (Data dependent) [1] | Extreme Jitter (Compounded) [7][18][19] |
| **Throughput** | > 100,000 readings/sec [9][24] | < 1,000 TPS (Public/Hybrid chains) [20] | Low (Ciphertext expansion) | Bottlenecked |
| **Scalability** | 1,000 - 10,000+ Devices [9][11] | Scalability Paradox [16][20] | Limited to small datasets [1][8] | 20 - 25 Nodes Max [7] |
| **Verdict** | Baseline | Fails Safety Standards | Fails Real-Time Req. | System Failure |

### 4.1 Analysis of the Collision

The matrix exposes the "Performance-Security Paradox". The most robust hybrid solutions, such as those proposed by Loukil et al. [7], provide excellent privacy and integrity but introduce latencies starting at 51 ms for extremely small networks (5 nodes) and degrading to 440 ms as the network scales to 50 nodes.

This is a critical finding. In a real-world factory requiring the coordination of thousands of sensors, a 440 ms latency

is not an inconvenience; it is a denial-of-service condition for the control loop. The reliance on blockchain consensus—even optimized "credit-based" Proof-of-Work—introduces probabilistic latency that fundamentally violates the deterministic requirements of the OT environment [3][5][11][17]. Furthermore, Fully Homomorphic Encryption (FHE) adds computational overhead 4–6 orders of magnitude higher than plaintext operations, rendering it infeasible for real-time edge processing [1][8].

Therefore, we conclude that Hybrid BC+HE architectures are architecturally invalid for Level 0-1 deployment.

## 5. THE GATEWAY ILLUSION: WHY OFFLOADING FAILS

A common counterargument in the literature suggests "offloading" heavy cryptographic operations from the resource-constrained edge (Level 1) to more powerful Supervisory Control (Level 2) gateways. The assumption is that SCADA systems can act as a buffer, aggregating sensor data before subjecting it to the heavy computational load of blockchain consensus or homomorphic encryption.

However, the analysis indicates this architectural pattern merely shifts the point of failure rather than resolving it. By moving the encryption boundary up to the SCADA layer, we create the Level 2-3 Inter-Layer Vulnerability, identified as the most critical gap in current research [15].

### 5.1 The SCADA Bottleneck

Level 2 SCADA systems are designed to aggregate high-velocity data from thousands of PLCs [9]. When these systems are tasked with performing homomorphic encryption or blockchain hashing on incoming data streams, they face a "computational versus encryption trade-off".

- **Throughput Saturation:** SCADA systems must process data for real-time monitoring. Introducing schemes like PrivDA (Privacy-Preserving Data Aggregation), while theoretically sound, introduces computational latency that scales poorly. Research shows that latencies rise significantly with just a handful of nodes [7].

- **The "Dead Zone":** This creates a security "dead zone" at the SCADA-to-Enterprise boundary [3]. This helps us arrive at idea that high-velocity operational data accumulates faster than the gateway can encrypt and commits it to a blockchain, forcing operators to choose between disabling security to maintain visibility or accepting telemetry-gap(reduced observability) to maintain security.

Existing solutions fail to bridge this gap. For instance, while Liang et al. [10] proposes a decentralized database for integrity, its confidentiality relies on heavy secret-sharing schemes that degrade real-time performance. Conversely, Loukil et al.'s [7] privacy-preserving aggregation is too slow for the volume of data at this specific boundary. The "Gateway" approach effectively turns the SCADA system into a bottleneck for the entire facility [7].

# 6. PROPOSED ARCHITECTURE: LIGHTWEIGHT TRUST AT THE EDGE

Given the "Determinism Barrier" at Level 0-1 and the "Bottleneck" at Level 2, this paper proposes that the industry must abandon the pursuit of "Hybrid BC+HE" for the physical edge. Instead, as detailed in Figure 2, we advocate for a Tiered Security Architecture that matches cryptographic complexity to available resources. This model strictly decouples the probabilistic enterprise layers from the deterministic edge.

## 6.1 Rejecting Blockchain at Level 0-1

The requirement for ≤ 10 ms deterministic response times renders blockchain consensus technically invalid for sensor-to-controller communications. The probabilistic nature of consensus cannot be reconciled with the safety-critical requirements of IEC 61508 [14].

## 6.2 The Lattice Alternative

Instead, this paper proposes the adoption of Lattice-Based Cryptography for the edge. To operationalize this framework within the "Deterministic Loop" defined in Section 3, we identify Module-LWE (ML-LWE) and N-th Degree Truncated Polynomial Ring Units (NTRU)-based primitives as the optimal classes for Level 0–1 deployment [25]. Unlike the computational heaviness of standard FHE or traditional RSA [8] —which we established introduces latencies exceeding 440 ms—these specific classes support Number Theoretic Transform (NTT) arithmetic. This algorithmic optimization accelerates processing by mapping polynomial operations into a frequency domain for rapid pointwise multiplication [26].

This optimization fundamentally alters system scalability by reducing complexity from the quadratic $O(n^2)$ growth of standard quadratic multiplication to linear-arithmetic $O(n \log n)$, ensuring that computational workload increases only marginally relative to data volume. Empirically, this enables cryptographic verification on standard edge controllers (e.g., ARM Cortex-M4) to execute in <0.5 ms , providing a safety margin of $20 \times$ against the strict $\leq 10$ ms WCET requirement by safety stadards in IEC 61508 [13]. This guarantees that security remains a background process, never colliding with the physics of the control loop.

## 6.3 Hardware Acceleration

To enable eventual integration with upper-layer homomorphic encryption, future research must focus on hardware acceleration (such as Field-Programmable Gate Arrays [FPGA] and Application-Specific Integrated Circuits [ASIC]) rather than algorithm optimization only. By embedding "lightweight" encryption primitives directly into silicon can we hope to bridge the performance gap identified in the Latency Collision Matrix.

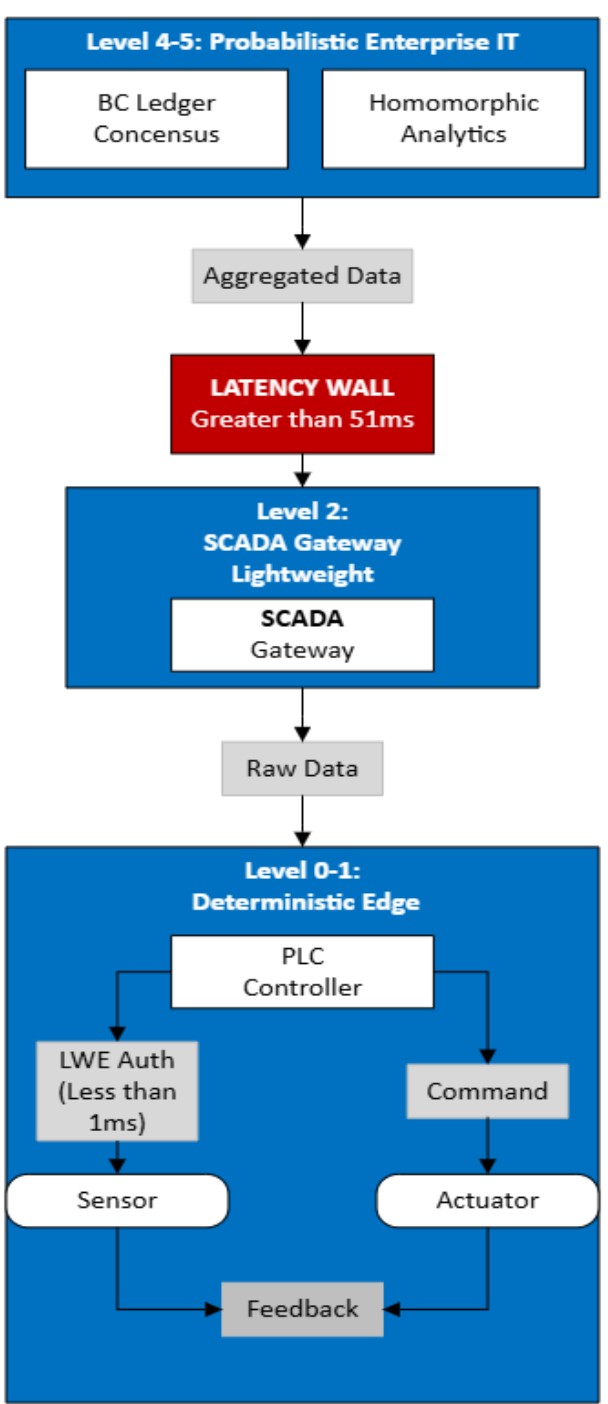

**Figure 2: Proposed architecture for offsetting resource limitations**

## 7. ACKNOWLEDGEMENTS

AI use attributions include 1)Cross-checking bibliography 2)Formula sanity check 3)Proofread- ACM compliance check 4)Proofread- Narrative flow 5)Proofread- Grammar 6) Proofread- Acronym First Occurrence 7)Bibliography formatting 8)Cross-section sample of assertions was checked to see if author provided evidence was directly supportive or triangulated(Corrective changes to narrative was made based on review outcome from AI which was fully human input)

## 8. CONCLUSION

This paper challenged the prevailing academic trend of applying hybrid Blockchain and Homomorphic Encryption architecture universally across Industrial Control Systems. By isolating the specific constraints of the "Edge" (Level 0-1), we demonstrated that the performance cost of these technologies—specifically latencies exceeding 51 ms—is fundamentally reconciled with the safety-critical requirements of IEC 61508 [13].

The "Performance-Security Paradox" is not merely an optimization hurdle; it is a barrier to entry for safety-critical deployment [9]. We conclude that while Blockchain and HE have significant utility in the upper enterprise layers (Level 4-5) for supply chain tracking and collaborative analytics, they are architecturally invalid for the physical control loop. To transition this work from purely  analytical modeling to empirically verifiable research, we would propose that future work will validate these latency arguments from existing research using Hardware-in-the-Loop (HIL) emulation[26]. A proposal would be to  stress-test the Latency Collision Matrix using a testbench consisting of Mininet-OT simulations coupled with physical PLC endpoints to quantify the exact jitter and latency introduced by blockchain integration.

Future research must pivot away from forcing these heavy protocols onto edge devices and instead focus on lightweight, lattice-based alternatives that secure the sensor without breaking the physics of the process.

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
