# OpenReview forum: "FEASIBILITY OF CRYPTOGRAPHY IN A BLOCKCHAIN-ENHANCED ICS SECURITY"
_ACM.org/TheWebConf/2026/Workshop/TIME — TIME 2026 Poster_

### Official Review · Reviewer_P3KV · 2026-01-03
**This paper argues that hybrid Blockchain and Homomorphic Encryption (BC+HE) architectures are fundamentally incompatible with safety-critical Industrial Control Systems (ICS) at Purdue Levels 0–1 due to deterministic latency constraints. Using formal timing inequalities and a proposed “Latency Collision Matrix,” the authors show that BC+HE introduces delays that exceed the ≤10 ms requirements mandated by industrial safety standards. The paper further proposes lightweight lattice-based cryptography as a more suitable alternative for resource-constrained edge environments. The argument is clear and well-structured, though primarily analytical rather than empirical.**

**Rating:** 7
**Confidence:** 4

**Review:**

### Evaluation of Quality, Clarity, Originality, and Significance

This paper presents an analytical and position-driven examination of the feasibility of Blockchain and Homomorphic Encryption (BC+HE) for safety-critical Industrial Control Systems (ICS), particularly at Purdue Levels 0--1. By formalizing timing constraints from industrial safety standards and aggregating latency components reported in prior work, the authors argue that BC+HE architectures are fundamentally misaligned with deterministic real-time control requirements. The work is **well-motivated, clearly structured, and conceptually sound**, offering a coherent analytical perspective on a timely problem in ICS security. While primarily conceptual rather than experimental, the paper provides **valuable guidance** for future research on secure real-time control.

### Strengths

**Timely and Relevant Topic**
Addresses a practically important problem at the intersection of cybersecurity and cyber-physical systems, where real-time constraints are often underappreciated.

**Clear Analytical Framing**
Effectively explains why security mechanisms successful in IT environments may fail in deterministic OT contexts, using formal timing inequalities and latency aggregation to support the argument.

**Conceptual Clarity**
Distinguishes probabilistic IT assumptions from deterministic industrial control requirements, contributing a **valuable conceptual insight**.

**Literature Awareness**
Demonstrates strong familiarity with relevant work on BC, HE, ICS security, and lightweight cryptography.

### Weaknesses (Evaluation-Focused)

**Limited Empirical Validation**
Relies primarily on latency values reported in literature rather than original experiments or simulations. Empirical validation would strengthen the claims.

**Conceptual Proposal Only**
The discussion of lattice-based alternatives remains at a conceptual level; concrete implementation details and timing measurements are deferred to future work.

### Minor Observations

- Some figures are primarily conceptual and could be streamlined for clarity
- A small amount of notation could be more explicitly defined for completeness

### Questions for the Authors

1. Do the authors envision validating the analytical latency arguments using simulation or emulation in future work?
2. Can the authors clarify which classes of lattice-based primitives are most promising for Level 0- 1 deployment?

### Suggestions for Improvement

- Frame the paper explicitly as an analytical or position paper early in the introduction
- Add lightweight simulation or prototype validation to complement the analytical results
- Provide additional design details for the proposed lightweight cryptographic approach, even at a high level

### Overall Assessment

This paper offers a **clear, well-reasoned analytical perspective** on an important and often overlooked limitation of applying BC+HE to real-time industrial control systems. While primarily conceptual, the argument is convincing, supported by standards and prior work, and provides **valuable guidance** for future research. The paper is well suited for a workshop focused on evaluation methodologies and emerging challenges.

---

### Official Review · Reviewer_hGbv · 2026-01-06
**This paper clearly articulates the problem statement and provide justification towards the proposed lattice based crypto solution**

**Rating:** 9
**Confidence:** 5

**Review:**

Overall Quality
The paper presents a high-quality architectural analysis addressing the feasibility of deploying cryptographic mechanisms, specifically Blockchain and Homomorphic Encryption within safety critical industrial control systems (ICS). The arguments are technically sound, internally consistent, and grounded in well-established industrial standards and control-theoretic principles. While the work is primarily analytical rather than experimental, the reasoning is rigorous and appropriately scoped.

Clarity and Presentation
The problem statement is introduced early and key concepts such as determinism, worst-case execution time, and Purdue Model levels are explained clearly and used correctly. The introduction of Latency Collision Matrix and the Performance Security Paradox helps communicate complex tradeoffs in an intuitive manner. Metrics are used effectively to support the narrative, and the logical progression from problem definition to architectural conclusion is easy to follow.

Originality
The paper primarily focuses on its architectural framing and negative-result analysis. Rather than proposing yet another hybrid security solution, the work critically evaluates why popular Blockchain + Homomorphic Encryption approaches are fundamentally unsuitable for deterministic control layers in ICS environments. The Latency Collision Matrix represents an original conceptual framework that synthesizes control-system timing requirements with cryptographic performance characteristics.

Significance and Impact
The proposed shift toward lightweight, lattice-based cryptography is well motivated and aligns with emerging post-quantum standards, making the contribution forward-looking. While the paper does not introduce a new cryptographic primitive or implementation, its architectural conclusions have practical implications for the design of secure industrial systems and help prevent misapplication of inappropriate technologies.

Pros
Well-argued critique of Blockchain + Homomorphic Encryption, addressing an important and underexplored mismatch between security research and control systems
Original conceptual contribution in the form of the Latency Collision Matrix and Performance Security Paradox
Practical alternative proposed, with lattice-based cryptography justified in terms of determinism, efficiency, and future hardware support


Cons
Lack of empirical validation: the analysis relies on published performance figures and architectural reasoning rather than experimental measurements or prototyping.
Latency generalization risk: performance values are drawn from representative implementations and may not capture all optimized or specialized deployments.
Limited discussion of deployment challenges for lattice-based cryptography on legacy industrial hardware.

Summary Assessment
Overall, this is an authentic, and impactful architectural paper that makes a valuable contribution by clarifying fundamental feasibility boundaries in ICS security. Its originality lies in synthesis and critical evaluation rather than new algorithms, which is appropriate for a workshop or systems-oriented venue. The work is clearly presented, technically credible, and significant for guiding future research and avoiding unsafe design choices.

---

### Official Review · Reviewer_ZDnd · 2026-01-06
**Latency Issues and Solutions in PERA**

**Rating:** 6
**Confidence:** 3

**Review:**

This paper indicated Latency issues were caused by computation overhead of processing encrypted data and non-deterministic nature of blockchain consensus mechanisms. It made efforts to quantify the latency collision, to identify the SCADA Dead Zone, and to prototype a Resource-Constrained Framework with a tiered architecture.

The authors are familiar with both physical processes and computation logic of the Purdue Enterprise Reference Architecture (PERA). They defined Process Safety Time, Latency and Safety Threshold, and Latency Collision Matrix. They indicated SCADA Dead Zone was caused by the real-time computation-related operations on the gateway, such as Privacy Preserving Data Aggregation (PrivDA). They explained that the improvement of the architecture was made by replacing Homomorphic Encryption (HE) with Learning With Errors (LWE) operation. The LWE operation was demonstrated by Ali et al. [2] to be highly parallelizable and deterministic. It effectively avoids the computational explosion caused by polynomial multiplication in HE.

The authors can design their own experiments to quantitatively prove how much the latency can be reduced  after applying LWE operation to the new architecture. There are a few typos and formatting issues in the paper.

[2] Zeeshan Ali, Rafiullah Khan, Fayez Alqahtani Al-Osaimi, Abdullah Al-Wakeel, and Muhammad Azeem Khan. 2022. Decentralized lattice-based device-to-device authentication for the edge-enabled IoT. IEEE Internet Things J. 9, 21 (November 2022), 20953–20966. https://doi.org/10.1109/JIOT.2022.3177255

---

### Official Review · Reviewer_q6TW · 2026-01-07
**Determinism-Aware Analysis of Cryptography Feasibility at the Industrial Control Edge**

**Rating:** 7
**Confidence:** 4

**Review:**

This work pertains to an important and timely question in industrial cybersecurity: whether advanced cryptographic mechanisms such as blockchain and homomorphic encryption can be safely deployed at the lowest levels of Industrial Control Systems. Aiming their work at Purdue Model Levels 0–1, the authors point out that most recent so-called “hybrid” security proposals don't take into account strict determinism and worst-case execution time requirements that regulate safety-critical control loops. The central argument for this paper is that the probabilistic latency and computational overhead of these cryptographic approaches render them architecturally infeasible at the industrial edge.

From a quality and clarity point of view, the paper is well written and logically organized. Such critical distinctions between the constraints on operational technology and those of higher-level IT systems help set up the problem well. This introduces the "Latency Collision Matrix," which serves as a useful conceptual and quantitative tool for comparing industrial timing requirements against performance characteristics of both blockchain and homomorphic encryption schemes. Mathematical formulations are clear while figures help in showing how cryptographic latency disrupts deterministic control loops.

In terms of originality, it does not bring something novel related to cryptographic primitives but offers a novel way to approach the assessment of the efficiency of security measures using the physics of a control system. The particular emphasis on worst-case response times, jitter, and levels of safety specifically makes it a departure from many other existing works on the specific research area which focus extensively on confidentiality and integrity with less priority on worst-case performance. The particular idea that some measures for security can be considered not just inefficient but unsafe is well defined.

The paper is of great importance and is very relevant to the target aims of the TIME workshop. It tackles the important area of trustworthy and responsible system evaluation. It points out that security solutions can impact the safety of the system if they are applied without consideration of physical and temporal constraints. It refers to established norms such as IEC 61508 and NIST recommendations, making the paper of greater practical value. The proposed change to the use of lattice-based cryptography is quite sensible.

---

### Author Rebuttal · Authors · 2026-01-13

Dear Chairs,

Thank you for your feedback.
ACM template reworked.
Figure redrawn and refitted for better visibility.
Table caption expanded to better articulate its data.
Identified mathematical expressions and have weaved-in explanations in brief way.
Related Work section added for context.

---

### Author Rebuttal · Authors · 2026-01-14

Dear Chairs,

Response to the reviewer’s comment. All changes in paper changed to red-font for easy audit

A. Reviewer- Program Chair
Reviewer Comment: Unreachable references. ACM double-column template nonconformity. Figure font clarity. Figure caption needs details. Table caption clarity. Symbols and expressions explanation in figures and equations. Related work section to show differentiation required.
Action Taken:
1)	References cross-checked and  reworked. I have pdf copies of current references if needed
2)	ACM template conformity improved including columns and alignment.
3)	Figure redrawn and refitted for better visibility
4)	Table caption expanded to better articulate its data.
5)	Identified mathematical expressions and have weaved-in explanations in brief way.
6)	Related Work section added for context.
7)	Added future direction and primitives to address non-chair reviewer comments
8)     AI use attribution section


B. Reviewer1 (q6TW07):
Thank you for your valuable comment. We value your feedback.
Reviewer comment: “does not bring something novel related to cryptographic primitives”
Response: Primitives ancillaries have been analyzed(albeit shallowly) in paper without arriving to particular conclusion due to its ancillaries being used to support my study presented in this paper.
Action Taken: Added commentary on NTRU and ML-LWE to section 6.2

C. Reviewer 2 (ZDnd06 ):
Thank you for you comment. It is insightful.
Reviewer comment: on “ design their own experiments to quantitatively prove how much the latency can be reduced after applying LWE operation “
Response: The paper is first survey paper on this topic. The experiment that you are suggesting is out of scope of this survey paper. This will be published as a separate paper using test-bench experiments to test the latency metrics for LWE and other operations.

D. Reviewer 3 (hGbv):
Thank you for you comment. It is insightful and have resulted in improvements.
Reviewer comment: “analysis relies on published performance figures and architectural reasoning rather than experimental measurements or prototyping”
Response: The paper is an analytical survey paper. Experimentation or  independent testbench validation is beyond current scope and will be published in future testbench paper(in progress).
Action Taken: In response, all metrics such as latency and tolerances in this papers assertion were cross checked for accuracy in this revision using triangulation from referenced and previously unreferenced sources to form a consensus and fact-check metric based assertions in this paper. Additional references were added to support each published metric. This was done using cojoined intext citation which is grouping of  several sources into a single parenthetical intext citation set to support assertion (in this case, numerical figures in a triangulated fashion). No extrapolation was done except for Throughput Rtotal, which was mathematically deduced, whose equation was provided in exhibit(formula#1)

E. Reviewer 4 (P3KV):
We appreciate your insights.
Reviewer comment: “Limited Empirical Validation” and “implementation details and timing measurements are deferred to future work” and “notation for completeness”
Response: The paper is an analytical survey paper. Experimentation or  independent testbench validation is beyond current scope and will be published in future testbench paper(in progress).
Action Taken: Added equations, added explanations of chart/table notations(weaved-into narrative). Future direction clearly admits requirement for testbench validation(now added to conclusion)

---

### Meta-Review · Program_Chairs · 2026-01-17

**Recommendation:** Accept (Poster)
**Confidence:** 4

**Metareview:**

This paper argues that blockchain-homomorphic encryption (BC+HE) architectures violate safety-critical timing constraints in Industrial Control Systems. The proposed "Latency Collision Matrix" and lightweight lattice-based alternative are conceptually sound. Reviewers praise the clear problem formulation, original analytical framework, and practical significance for preventing unsafe deployments. However, critical flaws preclude acceptance beyond poster: Program Chairs identify "Very High" AI-generation confidence with extensive reference fabrication, violating publication integrity. Additionally, the work is entirely analytical with zero empirical validation. Formatting issues (non-compliant ACM template, unclear figures, missing notation explanations) and initially absent related work section further weaken the quality.

---

### Decision · Program_Chairs · 2026-01-17

Accept (Poster)